# Effects of Magnetic Field and Ultrasound Applications on Endogenous Melatonin Content and Drought Stress Tolerance of Pepper Seedlings

Gökçen Yakupoğlu

Department of Horticulture, Faculty of Agriculture, Yozgat Bozok University, Yozgat 66100, Turkey; gokcen.yakupoglu@yobu.edu.tr

**Abstract:** Uncertainty about climate change exposes agriculture to high risks. Magnetic field (MF) applications are one of the methods that can be used to reduce the effects of environmental stress conditions. Melatonin (MEL) acts as a broad-spectrum antioxidant in eliminating the effects of damage caused in plants exposed to stress. This study aims to examine the effect of MF and ultrasound (US) applications on endogenous MEL levels in peppers and investigate the usability of treated seeds against drought stress. Pre-sowing pepper seeds, 0, 0.3, 0.9, 1.1 T MF and 0, 15, 30 min 40 Hz US were applied. The results show that the highest increase in MEL content was detected at 0.3 T MF with 82% and with 30 min of US application at 98%. MF and US treatments decreased the malondialdehyde (MDA) (19% and 35%, respectively) and hydrogen peroxide ($H_2O_2$) (52% and 58%, respectively) contents in seedlings. An increase of 24% and 22% (MF-US, respectively) was observed in catalase (CAT) enzyme activity with these applications. As a result, it was observed that MF and US treatments increased the endogenous MEL content and provided tolerance against drought stress. MF and US applications could be suggested as methods to increase drought tolerance in peppers by increasing the endogenous MEL content.

**Keywords:** pepper; magnetic field; ultrasound; endogenous melatonin; drought stress

## 1. Introduction

It can be observed that the use of land and water for agriculture by humans has not reached its peak yet; however, the growth in agricultural production has slowed down, the production capacity has been exhausted, and the environmental damage has increased [1]. Uncertainty about climate change exposes agriculture to high risks. The effects of climate change not only limit the unsustainable use of resources but also restrict rainfed and irrigation production practices. Future climate change scenarios indicate that planting patterns and management practices must be changed to adapt to these changes [2]. Approximately 1.2 billion people live in water-stressed areas where drought stress is intense, water is scarce and agricultural production is difficult. Pressures on water resources and fertile lands limit the agricultural production capacity. Drought and water scarcity endanger agricultural production, increasing poverty and malnutrition [1]. Agriculture is the sector that uses the most water (72%) in Turkey. The excessive use of water for agriculture has led to the drying up of water resources and drought is becoming more common in important agricultural areas of the country [3]. Various applications are used to increase the tolerance to drought or different abiotic stresses in agricultural production. Some of these applications are made to seeds and are called priming [4]. Plasma applications also form part of these applications.

A magnetic field (MF) is a vector quantity known as the force acting on electric charges in motion. Recently, many researchers in medicine, biology, and agriculture have been working on this subject [5]. Earth also naturally has an MF of 50 microteslas (mT) [6]. MF has positive and negative biological effects on living organisms. Studies have suggested

that MF has the potential to cause biological effects by affecting chemical reactions [7]. The stimulatory response induced by the seed application of MF is sustained until crop maturity as manifested by enhanced plant growth, leaf area, photosynthesis, biomass, and crop yield [8,9]. MF is an affordable and safe physical method that can be used to improve seed quality and viability, increase germination and seedling and plant growth, and cure deficiencies of seeds [10–12]. Different MF intensities and exposure times may cause different plant responses [13] and plant species may respond differently to MF. The seed preparation technique shows different responses according to the duration and frequency of exposure to MF, species, and seed characteristics [14]. Product efficiency decreases due to global climate change and MF applications could be used effectively to reduce the effects of adverse environmental conditions [15]. MF and plant growth regulators regulate plant growth by affecting the transport of metabolites and water and ions [16]. Magnetized water applied to the Moringa plant increased its tolerance against drought stress by increasing the wet–dry leaf weight, leaf area, improved stomatal conductivity, transpiration, and water use efficiency [17]. *Ficus carica* L. nodal explants exposed to drought stress (0, 3, and 6% polyethylene glycol 6000) were treated daily with MF for 0, 15, 30, and 60 min. The 15 min application gave the best response to drought stress. Short-term MF applications helped the plants absorb more water, increase leaf thickness and tolerate drought stress by accumulating more proline [18].

Melatonin (MEL) is a phytohormone synthesized from the aminoacid tryptophan (Trp). Trp is also the precursor of serotonin and indole 3 acetic acid (IAA) [19]. MEL acts as a broad-spectrum antioxidant in eliminating the effects of damage caused by the accumulation of free radicals in plants exposed to stress [20]. In addition, the consumption of products with high MEL content is important for health [21]. Biotechnological and molecular methods applied to plants can increase the level of endogenous MEL [22]. Studies have reported that MF applications affect the content of some endogenous phytohormones and increased endogenous gibberellic acid ($GA_3$) levels in pea seeds [23], mung bean [24] and tomato (70.2%) [25] were reported to be enhanced by MF applications. Although there are reports on the effects of MF applications on the synthesis of some endogenous hormones, no study has been found on their effect on the endogenous MEL levels. Since MEL is known as a broad-spectrum antioxidant that plays very important roles in a plant's response to various environmental stress factors, in this study, the possibility of whether MF applications can increase endogenous MEL levels is explored. It also investigated whether MF- and US-treated seeds could induce tolerance to drought stress with a change in endogenous MEL levels. Therefore, the current study aiming to examine the effect of MF and US applications on endogenous MEL levels in peppers and to investigate the usability of MF-treated seeds against drought stress was conducted.

## 2. Materials and Methods

### 2.1. Plant Material, MF and US applications

The experiment was carried out in glass greenhouses of Yozgat Bozok University Faculty of Agriculture. Long-fruited sweet pepper (*Capsicum annuum*) seeds (cv. Çetinel) were used in the experiment. MF and US applications were carried out in the following two ways:

1. With the help of Hall Effect equipment (HEMS-5405 Dipole-GMW, San Carlos, CA, USA) (Figure S1), an MF of 0, 0.3, 0.9, and 1.1 Tesla was applied to dried pepper seeds for 5 min.
2. With an ultrasonic bath (Elma-Elmasonic S, Singen, Germany), 40 Hz US was applied to dry pepper seeds at room temperature for 0, 15 and 30 min.

MF- and US-treated seeds were grown in 1:1 peat and perlite medium in seedling trays in unheated greenhouses. When the plants reached the 4-leaf stage, the trays were irrigated up to the field capacity, and then the irrigation was ceased. Half of the seedlings were exposed to drought stress for 12 days, while the other half were watered regularly. Following drought stress, various biochemical analyses were conducted on seedling leaves.

*2.2. Sampling and Methods*

2.2.1. MEL Content

A modified version of the method reported in Arnao and Hernández-Ruiz's work [26] was used for sample extraction. Tubes containing 3 mL of chloroform and 0.5 g of frozen plant tissue were placed in a shaker at 4 °C in the dark for 17 h. The tubes were centrifuged for 20 min at 6000× *g* and 4 °C, and the resulting supernatant was transferred to another tube. The remaining plant residue in the tubes was washed with 0.5 mL of chloroform. The liquid part of sample was evaporated using a CentriVap vacuum concentrator (Labconco, Kansas City, MO, USA) attached to a refrigerated CentriVap vapor trap. The residue was redissolved in 0.5 mL of methanol, filtered with a 0.45 μm filter, and used for analysis. An HPLC system consisting of an Intersil ODS-2 column (GL Sciences, 5 μm, 150 × 4.6 mm) equipped with Prominance UFLC equipment (Shimadzu, Kyoto, Japan) and an RF-20A fluorescence detector was used for the measurement of melatonin. Melatonin contents in each sample were calculated by comparing the sample peak area with the standard melatonin curves. The values were expressed as ng g$^{-1}$ fresh weight (ng g$^{-1}$ FW) of tissues.

2.2.2. Chlorophyll Content

The leaf samples taken in 4 replications were filtered through filter paper (Whatman No: 42) after homogenization with 80% acetone solution. After reading the absorbance at 645 and 663 nm using a spectrophotometer (Shimadzu UV-1800, Kyoto, Japan), the concentrations of chlorophyll pigments in the filtered extract solution were calculated according to the formula given by Güneş et al. [27].

2.2.3. Carotenoid Content

Carotenoid analysis was performed according to the method outlined by Witham [28]. The leaf samples were extracted with 80% acetone filtered through filter paper. Absorbance readings were taken using the spectrophotometer at 480, 645, and 663 wavelengths. Carotenoid amounts were calculated as mg g$^{-1}$ FW.

2.2.4. Malondialdehyde (MDA) Analysis

According to the method given by Zhang et al. [29], the MDA content was determined. Leaf samples taken randomly from plants were extracted with 0.1% TCA. The extracts were centrifuged, 1 mL was taken from the supernatant in the tube, and 0.5% TBA (thiobarbituric acid) containing 20% TCA was added. Absorbance of the samples was read at 450, 532, and 600 nM.

2.2.5. Total Phenolic Substance

Total phenolic substance analysis was carried out according to the method outlined by Ahmed et al. [30] using the Folin–Ciocalteu colorimetric method. The % sodium carbonate ($Na_2CO_3$) solution was made up. Leaves were crushed with 95% ethanol. The Folin–Ciocalteu reagent and $Na_2CO_3$ were added to the sample extracts and kept in the dark. Spectrophotometer readings were performed at a wavelength of 765 nm. Total phenolic substance amounts are given in mg/g as gallic acid (GAE).

2.2.6. Proline Analysis

Quantification of proline was carried out according to the method described by Bates et al. [31]. Accordingly, the leaf samples were homogenized with 3% sulfosalicylic acid, acid ninhydrin, and glacial acetic acid added to the filtered homogenate and kept in a water bath at 100 °C for 1 h. After that, the samples were extracted with toluene and the absorbance was read using the spectrophotometer at 520 nm. The amount of proline was determined as μmol proline/g fresh weight.

### 2.2.7. Hydrogen Peroxide ($H_2O_2$) Analysis

A modified version of the method specified in the work of Özden et al. [32] was used to determine the $H_2O_2$ content. Fresh leaf samples (0.25 g) were extracted with 0.1% TCA (trichloroacetic acid). The samples were centrifuged at $10,000 \times g$ for 15 min, and later, their absorbance values were read at 390 nm using the spectrophotometer by adding 100 µL of the supernatant, 100 µL of K-P buffer solution, and 1.5 mL of KI solution.

### 2.2.8. Enzyme Analysis

Enzymatic extraction was performed following the method of Seckin et al. [33] and the total soluble protein content of the extract was determined using bovine serum albumin (BSA) as a standard [34]. For extraction, 0.25 g of the leaf samples was homogenized by crushing it with 1.5 mL of 50 mM phosphate buffer solution (pH 7.8) containing 1 mM Na-EDTA and 2% PVPP (*w/v*) on ice. The homogenized samples were centrifuged at $14,000 \times g$ at 4 °C for 30 min, and then the resulting supernatant was used for analysis. The activities of peroxidase (POX, E.C. 1.11.1.7) and catalase (CAT, E.C. 1.11.1.6) were measured by following the methods published by Dolatabadian et al. [35] and Bergmeyer [36], respectively. Measurements were performed in three replications and enzyme activity was expressed in units of $mg^{-1}$ protein.

### 2.3. Statistical Analysis

The experiment was set up with three replications (12 plants each) according to the factorial (two factor) experimental design. For calculating the averages of all the data obtained during the research, "Microsoft Office XP EXCEL" was used. The data were analyzed using the SPSS 20.0 package program and the effect of treatments on the means of the investigated characteristics was determined by Duncan's Multiple Range Test at the $p = 0.05$ level.

## 3. Results

### 3.1. The Effect of Magnetic Field (Tesla) Applications

It was observed that MF applications of different intensities to the seeds increased the endogenous MEL content of the seedlings (Table 1). MEL content, which was found as 84.38 ng $g^{-1}$ FW in the seedlings obtained from untreated seeds, increased with MF applications and the highest MEL content was determined in 0.3 T applied seeds with 153.23 ng $g^{-1}$ FW. The endogenous MEL content also increased in the seedlings exposed to drought stress. When the interaction effect of the factors was examined, although it is not statistically significant, the lowest MEL content was determined in the 0T-C application with 62.28 ng $g^{-1}$ FW, and the highest in the 0.3 T-D application with 184.47 ng $g^{-1}$ FW. When the effects of MF applications on CAT activity were examined, the lowest enzyme activity was determined in the 0 T (1.04 U $mg^{-1}$ protein) application. MF applications increased the antioxidant enzyme activities. The highest enzyme activity was determined in the application of 1.1 T (1.29 U $mg^{-1}$ protein). The main effect of stress was significant and CAT enzyme activity increased with drought stress. The interaction of stress and application also significantly affected CAT activity and the highest activity was found in the application of 0.9 T-D (1.44 U $mg^{-1}$ protein). When the POX content, which is one of the antioxidant enzymes, was examined, the lowest activity was found in the 1.1 T (0.64 U $mg^{-1}$ protein) application under non-stress conditions, while the highest enzyme activity was determined in the 1.1 T (1.66 U $mg^{-1}$ protein) application under drought stress.

When the malondialdehyde (MDA) content, which is one of the degradation products of the cell membranes, is examined, it was determined that the MF applications reduced the membrane deterioration. Maximum deterioration was observed in the 0 T application with 0.77 µmol $g^{-1}$ FW, while minimum deterioration was reported in the 0.9 T application with 0.62 µmol $g^{-1}$ FW. As expected, the maximum amount of MDA detected with drought stress was 0.86 µmol $g^{-1}$ FW. The interaction effect of the factors was also significant and the highest MDA content was found in 0 T-D, and 1.1 T-D applications with 0.96 µmol $g^{-1}$

FW, and the lowest in 1.1 T-C applications with 0.46 μmol g$^{-1}$ FW. All MF applications reduced the proline content and the highest amount of proline was detected as the 0 T application with 0.19 μmol g$^{-1}$ FW. It was determined that the amount of proline increased with drought stress application. MF applications significantly affected the amount of $H_2O_2$ content, which plays an important role in oxidative stress; its amount was reduced in all MF applications compared to the 0 T application. Drought also increased the $H_2O_2$ content of the seedlings and among the applications, the highest $H_2O_2$ accumulation was found in the 0 T-D application with 0.49 μmol g$^{-1}$ FW, while it was determined that the $H_2O_2$ accumulation decreased in MF applications in drought conditions.

**Table 1.** Effects of MF application on MEL, CAT, POX, MDA, proline and $H_2O_2$ contents of pepper seedlings.

| Treatments | | MEL ng g$^{-1}$ FW | CAT (U mg$^{-1}$ Protein) | POX (U mg$^{-1}$ Protein) | MDA μmol g$^{-1}$ FW | Prolin μmol g$^{-1}$ FW | $H_2O_2$ μmol g$^{-1}$ FW |
|---|---|---|---|---|---|---|---|
| Tesla | | | | | | | |
| 0 T | | 84.38 [d] | 1.04 [b] | 1.05 [ab] | 0.77 [a] | 0.19 [a] | 0.31 [a] |
| 0.3 T | | 153.23 [a] | 1.09 [b] | 0.81 [c] | 0.72 [a] | 0.15 [b] | 0.15 [c] |
| 0.9 T | | 100.54 [c] | 1.19 [ab] | 0.98 [b] | 0.62 [b] | 0.13 [b] | 0.23 [b] |
| 1.1 T | | 117.30 [b] | 1.29 [a] | 1.15 [a] | 0.71 [a] | 0.16 [b] | 0.19 [b] |
| Stress | | | | | | | |
| Control (C) | | 84.50 [b] | 1.05 [b] | 0.73 [b] | 0.55 [b] | 0.08 [b] | 0.16 [b] |
| Drought (D) | | 143.14 [a] | 1.26 [a] | 1.26 [a] | 0.86 [a] | 0.22 [a] | 0.28 [a] |
| Tesla × Stress | | | | | | | |
| Control | 0 T | 62.28 | 0.97 [d] | 0.87 [cd] | 0.57 [c] | 0.16 [b] | 0.12 [d] |
| | 0.3 T | 121.98 | 1.04 [cd] | 0.69 [de] | 0.69 [b] | 0.08 [c] | 0.14 [d] |
| | 0.9 T | 67.63 | 0.94 [d] | 0.74 [cde] | 0.48 [c] | 0.04 [d] | 0.21 [bc] |
| | 1.1 T | 86.43 | 1.26 [abc] | 0.64 [e] | 0.46 [c] | 0.05 [d] | 0.16 [cd] |
| Drought | 0 T | 106.47 | 1.11 [bcd] | 1.22 [b] | 0.96 [a] | 0.21 [a] | 0.49 [a] |
| | 0.3 T | 184.47 | 1.14 [bcd] | 0.93 [c] | 0.74 [b] | 0.21 [a] | 0.16 [cd] |
| | 0.9 T | 133.45 | 1.44 [a] | 1.21 [b] | 0.76 [b] | 0.22 [a] | 0.24 [b] |
| | 1.1 T | 148.17 | 1.32 [ab] | 1.66 [a] | 0.96 [a] | 0.24 [a] | 0.22 [b] |
| Significance | | | | | | | |
| Tesla | | ** | ** | *** | * | ** | ** |
| Stress | | ** | * | *** | ** | ** | ** |
| Tesla × Stress | | ns | * | *** | ** | ** | ** |

ns, *, **, ***, not significant, significant at *p* < 0.05, 0.01 or 0.001, respectively. Control indicates plants not exposed to drought stress conditions. The difference between the means in the same column starting with the same letter is not statistically significant according to Duncan's test (*p* < 0.05).

The effects of MF applications and drought stress on chlorophyll, carotenoid contents and total phenolics are presented in Table 2. When the chlorophyll content was examined, only MF applications statistically significantly affected the chlorophyll content and the highest chlorophyll content was determined in the 0.9-T application with 255.57 mg g$^{-1}$ FW. Stress conditions were found to be statistically significant in terms of carotenoid content, and the carotenoids content of drought-stressed plants were significantly higher than those of non-stressed plants. It was determined that MF applications slightly increased the total phenolic content, and as the application intensity increased, the total phenolic content decreased. The highest phenolic content was found for the 0.3 T-C application with 8.91 mg g$^{-1}$ FW (Table 2).

**Table 2.** Effects of MF application on seeds on chlorophyll, carotenoid and total phenolic contents of pepper seedlings.

| Treatments | | Chlorophyll Content mg g$^{-1}$ FW | Carotenoid Content mg g$^{-1}$ FW | Total Phenolic GAE mg g$^{-1}$ FW |
|---|---|---|---|---|
| Tesla | | | | |
| 0 T | | 248.99 [ab] | 9.69 | 7.84 [ab] |
| 0.3 T | | 221.07 [b] | 8.81 | 8.14 [a] |
| 0.9 T | | 255.57 [a] | 9.57 | 7.00 [bc] |
| 1.1 T | | 246.48 [ab] | 9.29 | 6.54 [c] |
| Stress | | | | |
| Control (C) | | 216.47 | 8.20 [b] | 7.27 |
| Drought (D) | | 269.59 | 10.39 [a] | 7.49 |
| Tesla × Stress | | | | |
| Control | 0 T | 216.79 | 8.29 | 7.08 [c] |
| | 0.3 T | 202.72 | 7.77 | 8.91 [a] |
| | 0.9 T | 228.25 | 8.76 | 6.57 [c] |
| | 1.1 T | 218.13 | 7.97 | 6.51 [c] |
| | 0 T | 281.19 | 10.69 | 8.59 [ab] |
| Drought | 0.3 T | 239.42 | 9.86 | 7.36 [bc] |
| | 0.9 T | 282.89 | 10.40 | 7.42 [bc] |
| | 1.1 T | 274.84 | 10.60 | 6.57 [c] |
| Significance | | | | |
| Tesla | | ** | ns | ** |
| Stress | | ns | ** | ns |
| Tesla × Stress | | ns | ns | ** |

ns, **, not significant, significant at *p* < 0.01. Control indicates plants not exposed to drought stress conditions. The difference between the means in the same column starting with the same letter is not statistically significant according to Duncan's test (*p* < 0.05).

### 3.2. The Effect of US Applications

When the endogenous MEL content of the seedlings obtained from the seeds that were treated with US at different minutes was examined, a higher MEL content was detected in the seedlings that were raised from US-treated seeds for increasing durations (Table 3). The lowest MEL content (84.37 ng g$^{-1}$ FW) was detected in untreated seedlings, and the highest (167.27 ng g$^{-1}$ FW) MEL content in US treated seedlings for 30 min. The endogenous MEL content in drought conditions was higher than that in the control environments. Among the applications, the lowest MEL content was determined in the 0 min C (US not applicated) application with 62.28 ng g$^{-1}$ FW, and the highest in the 30 min D with 183.89 ng g$^{-1}$ FW application. When the effect of US applications made at different times on CAT, one of the antioxidant enzymes, was examined, it can be observed that the US application increased the enzyme activity. When the effects of the applications were examined, the lowest enzyme activity was determined in the 0 min C application with 0.97 U mg$^{-1}$ protein, and the highest activity was determined in the 15 min D application with 1.33 U mg$^{-1}$ protein. It was also observed that drought application increased POX enzyme activity.

The effects of US applications on MDA accumulation were found to be significant and the amount of MDA detected in the 0 min application (0.77 µmol g$^{-1}$ FW) decreased significantly (0.50 µmol g$^{-1}$ FW) following the 15 min application. The MDA content was found to be the highest in plants under drought stress. In US applications, the highest MDA with 0.96 µmol g$^{-1}$ FW was in the 0 min D treatment and the lowest with 0.40 µmol g$^{-1}$ FW was in the 15 min C treatment. Similar to the MF application, the amount of proline in the US application decreased with the US applications. While the amount of proline increased in plants under drought stress, the highest proline content was detected in the 15 min D application with 0.25 µmol g$^{-1}$ FW. It was also observed that the amount of $H_2O_2$

$\mu$mol g$^{-1}$ FW decreased in US-treated plants. It was found that drought stress increased the amount of H$_2$O$_2$ and US applications significantly reduced the accumulation of H$_2$O$_2$.

**Table 3.** Effects of seeds on US application on MEL, CAT, POX, MDA, proline and H$_2$O$_2$ contents of pepper seedlings.

| Treatments | | MEL ng g$^{-1}$ FW | CAT (U mg$^{-1}$ Protein) | POX (U mg$^{-1}$ Protein) | MDA $\mu$mol g$^{-1}$ FW | Prolin $\mu$mol g$^{-1}$ FW | H$_2$O$_2$ $\mu$mol g$^{-1}$ FW |
|---|---|---|---|---|---|---|---|
| 40 Hz | | | | | | | |
| 0 min | | 84.37 [c] | 1.04 [b] | 1.05 | 0.77 [a] | 0.19 [a] | 0.31 [a] |
| 15 min | | 109.63 [b] | 1.27 [a] | 1.03 | 0.50 [c] | 0.17 [b] | 0.28 [a] |
| 30 min | | 167.27 [a] | 1.27 [a] | 0.91 | 0.61 [b] | 0.14 [c] | 0.13 [b] |
| Stress | | | | | | | |
| Control (C) | | 95.93 [b] | 1.14 | 0.83 [b] | 0.48 [b] | 0.09 [b] | 0.11 [b] |
| Drought (D) | | 144.92 [a] | 1.24 | 1.15 [a] | 0.76 [a] | 0.23 [a] | 0.36 [a] |
| Hz × Stress | | | | | | | |
| Control | 0 min | 62.28 [d] | 0.97 [c] | 0.87 | 0.57 [cd] | 0.16 [d] | 0.12 [cd] |
| | 15 min | 74.87 [d] | 1.21 [ab] | 0.82 | 0.40 [e] | 0.08 [e] | 0.15 [c] |
| | 30 min | 150.64 [b] | 1.25 [ab] | 0.81 | 0.48 [de] | 0.04 [f] | 0.06 [d] |
| Drought | 0 min | 106.47 [c] | 1.11 [bc] | 1.22 | 0.96 [a] | 0.21 [c] | 0.49 [a] |
| | 15 min | 144.39 [b] | 1.33 [a] | 1.24 | 0.60 [c] | 0.25 [a] | 0.41 [b] |
| | 30 min | 183.89 [a] | 1.28 [ab] | 1.00 | 0.73 [b] | 0.23 [b] | 0.19 [c] |
| Significance | | | | | | | |
| Hz | | ** | ** | ns | ** | ** | ** |
| Stress | | ** | ns | ** | ** | ** | ** |
| Hz × Stress | | * | ** | ns | * | ** | ** |

ns, *, **, not significant, significant at *p* < 0.05, 0.01, respectively. Control indicates plants not exposed to drought stress conditions. The difference between the means in the same column starting with the same letter is not statistically significant according to Duncan's test (*p* < 0.05).

When the chlorophyll and carotenoid contents of the US applications were examined, only the stress conditions were statistically found to be significant, and the highest chlorophyll and carotenoid contents (278.14 mg g$^{-1}$ FW–10.98 mg g$^{-1}$ FW, respectively) were detected in the drought application (Table 4). On the other hand, US applications significantly increased the phenolic content, and the highest phenolic content detected in the application of 30 min C with 9.30 mg g$^{-1}$ FW.

**Table 4.** Effects of US application on seeds on chlorophyll, carotenoid, total phenolic contents of pepper seedlings.

| Treatments | Chlorophyll Content mg g$^{-1}$ FW | Carotenoid Content mg g$^{-1}$ FW | Total Phenolic GAE mg g$^{-1}$ FW |
|---|---|---|---|
| 40 Hz | | | |
| 0 min | 248.99 | 9.49 | 7.84 [b] |
| 15 min | 248.96 | 9.89 | 6.61 [c] |
| 30 min | 246.58 | 9.63 | 8.56 [a] |
| Stress | | | |
| Control (C) | 224.22 [b] | 8.36 [b] | 7.54 |
| Drought (D) | 278.14 [a] | 10.98 [a] | 7.93 |
| Hz × Stress | | | |

**Table 4.** *Cont.*

| Treatments | | Chlorophyll Content mg g$^{-1}$ FW | Carotenoid Content mg g$^{-1}$ FW | Total Phenolic GAE mg g$^{-1}$ FW |
|---|---|---|---|---|
| Control | 0 min | 216.79 | 8.29 | 7.08 [cd] |
| | 15 min | 223.87 | 8.46 | 6.24 [d] |
| | 30 min | 213.99 | 8.32 | 9.30 [a] |
| Drought | 0 min | 281.19 | 10.69 | 8.59 [ab] |
| | 15 min | 274.04 | 11.32 | 6.97 [cd] |
| | 30 min | 279.18 | 10.94 | 7.82 [bc] |
| Significance | | | | |
| Hz | | ns | ns | ** |
| Stress | | ** | ** | ns |
| Hz × Stress | | ns | ns | ** |

ns, **, not significant, significant at $p < 0.01$. Control indicates plants not exposed to drought stress conditions. The difference between the means in the same column starting with the same letter is not statistically significant according to Duncan's test ($p < 0.05$).

## 4. Discussion

MF applications to the seed, also called magnetopriming, involve a dry seed preparation process that increases the germination rate and seedling strength of the products and does not harm the seeds. Applying MF to seeds is a very popular method in the agricultural sector [25,37–39]. It is known that MF application in plants and animals is used to increase resistance to adverse environmental conditions and diseases. MF increases seed germination and growth by increasing water and nutrient intake, protein, carbohydrate, photosynthesis and enzyme activities. It also reduces the negative effects of abiotic stress factors such as drought and salinity [15]. When other priming (osmo, hydro etc.) applications are carried out to the seeds, the seeds are moistened, so their storage time is reduced, and it is more convenient to plant them immediately after the application. When MF is applied, it can be an alternative to increase seed viability during the storage period [40]. There are not many studies on the effect of MF and US applications on the content of phytohormones [21,23,41]. To the best of our knowledge, no study was found on the effect of MF applications on the endogenous MEL content in plants; therefore, this is the first study to report that seed MF applications enhanced tolerance to drought stress at the seedling stage.

MF and US applications significantly affected the endogenous MEL content of pepper seedlings and higher MEL levels were detected in those who were raised from seeds treated with MF and US. Similar to our study, it was found that the content of endogenous IAA and cytokinin levels in pea seeds increased with plasma application with different durations (0, 120 and 600 s). It has been stated that increased cytokinin and auxin levels can stimulate cell elongation, division and proliferation. However, the authors noted that the mechanism behind the increased germination was not fully understood [23]. MEL is a biostimulating molecule that is considered to be directly involved in regulating many primarily physiological processes from seed germination to fruit maturation [20]. In a study conducted to determine the effect of endogenous MEL content on germination under chilling stress, it was reported that the varieties with high seeds MEL contents had higher germination percentages and rates [42]. It was determined that genotypes of Arabidopsis and tomato with high endogenous MEL contents were more tolerant to *Botrytis cinerea* infection and cadmium stress, respectively [43,44]. Similarly, cold plasma (a high-frequency (40 KHz), high-voltage generator (5 kV)) application applied to mung beans increased the amount of endogenous GA$_3$ by ~2.8 times, resulting in improved seed germination and seedling root growth. The change in GA$_3$ content was also found in parallel with radicle growth [24]. Podleśny et al. [41] reported that 30 and 85 mT MF applications to pea seeds significantly increased the amount of IAA and GA$_3$ in seeds, shoots and roots. Similarly, Selim et al. [45] stated that when they germinated tomato, wheat and

pea seeds with MF-treated water, the amount of GA, IAA and cytokinins (Kinetin, Zeatin) increased, while the amount of abscisic acid decreased. In a study where fig (*Ficus carica* L.) explants were treated with MF with varying durations (0, 15, 30, and 60 min), short-term (15 min) MF applications improved the drought stress tolerance of explants not only by increasing water uptake under severe stress conditions, but also by increasing leaf thickness and inducing proline accumulation in the leaves [18]. MF treatment promoted cambium differentiation, photosynthesis, stomatal conductivity, water absorption, and nutrient uptake in plants suffering from drought, while decreasing ROS generation in plants under stress. MF treatment regulates photosynthesis, stomatal conductivity and transpiration in plants exposed to heat and light stress [15].

Recently, MEL has been identified as a phytohormone [20]. According to reports, it is present in all plant species and regulates the circadian rhythm [46]. Aspects including germination, growth, root development, fruiting, parthenocarpy, maturity, and post-harvest have all been associated with phytomelatonin. MEL had a role as a protective and alleviating agent against both biotic and abiotic stress factors [20]. Endogenous MEL levels increase in plants to enhance tolerance to environmental stressors [22]. As a result of butafenacil and cadmium-induced oxidative stress in rice (*Oryza sativa* L.) seedlings, the activities of intermediate enzymes such as T5H, TDC and HIOMT, which are involved in melatonin biosynthesis, increased, resulting in significant increases in endogenous MEL content [47,48]. Similarly, in our study, the endogenous MEL content was higher in treatments exposed to drought stress. In a study that investigated the effects of endogenous MEL deficiency caused by silencing the caffeic acid o-methyltransferase 1 (COMT1) gene involved in MEL biosynthesis against high-temperature stress, it was observed that MEL-deficient tomato plants showed more severe symptoms (increased MDA levels, membrane damage, and oxidized and insoluble protein accumulation) than the wild-type plants. As a result of the study, it has been reported that optimal endogenous MEL levels are necessary to cope with stress conditions [49]. In our study, similar to the endogenous MEL content, CAT enzyme activities increased with MF applications, and the highest CAT activity was detected in the 1.1 T application. On the other hand, POX activity decreased with MF application. Similar to MF application, US applications increased CAT contents while decreasing the POX activity. It has been reported that the MDA content decreased and antioxidant enzyme activities increased in *Celosia argentea* plants treated with MF for Cd phytoremediation responses under drought stress. The results showed that magnetic field application alleviated the detrimental effects of drought stress, increasing the levels of photosynthetic pigments, transpiration rate, and antioxidant enzyme activity in plant tissues [50]. Our investigation also found that MF and US treatments decreased the amounts of MDA, proline, and $H_2O_2$ in seedlings. Chen et al. [51] reported that MF application in mung beans reduced the toxic effects of cadmium stress and decreased MDA, $H_2O_2$ and EC contents. Korkmaz et al. [42] studied the effect of endogenous MEL contents on chilling stress and found that seedlings raised from seeds with higher endogenous MEL contents had lower MDA and $H_2O_2$ contents under shilling conditions. However, they stated that an optimum MEL level is required for effective enzymatic activity against adverse environmental conditions. It was concluded that the effects of MF application depend on exposure time, vigor and plant species, and MF has the potential to increase crop production (such as photosynthesis, respiration, nutrient uptake, plant metabolism, enzyme and protein production) by regulating biochemical and physiological processes. A specific feature of MF is by increasing the high production of antioxidants, which can alleviate the negative effects of various stress factors [52]. Gene ontology and transcriptome studies indicated that 85% of the total genes significantly down-regulated by MF applied parallel to root growth compared to untreated roots are concentrated in plastid biological activities such as metabolism and chloroplast formation [53]. According to the results of our study, MF and US applications to pepper seeds improved antioxidant enzyme activities by increasing their endogenous MEL content, reducing cellular damage, and increasing tolerance to drought stress.

## 5. Conclusions

Few studies have reported the positive effects of MF treatment under environmental stress conditions by affecting phytohormones in plants. The result of this study revealed that MF and US treatments increased the endogenous MEL content of pepper seedlings which, in turn, provided tolerance against drought stress. Boosted drought stress tolerance could be attributed to elevated antioxidant enzyme activity and reduced oxidative damage. Therefore, MF and US applications can be recommended as a cheap, fast, practical and chemical-free method for increasing endogenous MEL content and drought tolerance. In future studies, different durations and intensities could be investigated to determine the effects of MF and US applications on different species. Moreover, it can be examined how the quality and content of products obtained from MF or US-treated plants that are used in human nutrition change.

**Supplementary Materials:** The following supporting information can be downloaded at: https://www.mdpi.com/article/10.3390/horticulturae9060704/s1, Figure S1: Hall Effect equipment (HEMS).

**Funding:** This research received no external funding.

**Data Availability Statement:** The data presented in this study are available upon request from the corresponding author.

**Conflicts of Interest:** The author declares no conflict of interest.

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
