# Peer review of "Effects of Magnetic Field and Ultrasound Applications on Endogenous Melatonin Content and Drought Stress Tolerance of Pepper Seedlings"

_horticulturae, doi:10.3390/horticulturae9060704_

Round 1

Reviewer 1 Report

I have reviewed this manuscript entitled " Effects of Magnetic Field and Ultrasound Applications on Endogenous Melatonin Content and Drought Stress Tolerance of Pepper Seedlings ". The topic is interesting to examine the effect of MF and ultrasound applications on endogenous MEL levels in pepper and to investigate the usability of MF-treated seeds against drought stress. However, the manuscript can be further considered for recommendation after the following a few comments and concerns are addressed.

1.      Improve the introduction part.

2.      State the novelty of this work at the end of introduction.

3.     In Materials and Methods section: Please describe the extraction method used to obtain the extract samples after the treatment.

4.     Could you add the chromatogram of HPLC analysis with the concerned picks (Melatonin pick…).

5.     The data must be improved by antioxidant test (DPPH Test…) to ensure that this treatment does not affect the antioxidant potential.

6.     The conclusion section could be condensed to a take home message based on the findings in the study. Now a large part is also a summary which could be found in the  abstract as well

Please resubmit the work with these considerations and the research paper will be fully evaluated.

 Best regards.

Author Response

Dear Reviewer

I would like to thank you for the time and effort you put in evaluating this manuscript. The comments from you were very helpful. I have made the following changes and corrections (indicated in red throughout the text) on the revised manuscript.

I hope that this revised manuscript serves the purpose of your satisfaction. However, if anything else needs to be done, I would like to let you know that I am willing to cooperate. I really appreciate your help.

Comments and Suggestions for Authors

I have reviewed this manuscript entitled " Effects of Magnetic Field and Ultrasound Applications on Endogenous Melatonin Content and Drought Stress Tolerance of Pepper Seedlings ". The topic is interesting to examine the effect of MF and ultrasound applications on endogenous MEL levels in pepper and to investigate the usability of MF-treated seeds against drought stress. However, the manuscript can be further considered for recommendation after the following a few comments and concerns are addressed.

  1. Improve the introduction part.

I have improved this part by adding drought information and the hypothesis behind this research to the last paragraph.

  1. State the novelty of this work at the end of introduction.

Since MEL is known as a broad-spectrum antioxidant that plays very important roles in plant’s response to various environmental stress factors, in this study, the possibility of whether MF applications can increase endogenous MEL levels is explored. It also investigated whether MF and US-treated seeds could induce tolerance to drought stress with a change in endogenous MEL levels. Added.

  1. In Materials and Methods section: Please describe the extraction method used to obtain the extract samples after the treatment.

The methods of extraction was added.

  1. Could you add the chromatogram of HPLC analysis with the concerned picks (Melatonin pick…).

Yes, the chromatogram could be given, but I did not add it for the sake of being succinct. Also, I have not come across the chromatogram peak given very often in the other articles I followed.

  1. The data must be improved by antioxidant test (DPPH Test…) to ensure that this treatment does not affect the antioxidant potential.

Unfortunately, our research has been terminated and we do not have enough samples to conduct additional analyses. Therefore, we do not have the opportunity to do this analysis. However, we thank the referee for his/her suggestion and we will consider it for the future studies.

  1. The conclusion section could be condensed to a take home message based on the findings in the study. Now a large part is also a summary which could be found in the abstract as well

The conclusion part has been improved according to referee’s suggestion.

Reviewer 2 Report

Line 37-38; (drought in 2008), please update the information as too old for now.

line 80; please determine the full scientific name and variety of pepper.

line 99; how did you prepare the extraction for chemical analysis please explain?

In all tables; in some values, you did not mention standard deviations, and have not shown the letter in superscripts.

why did you use four different controls? it was not described in the methodology.

Conclusion; nothing here is about the conclusion. You just recommended for future studies, please reshape your conclusion and conclude your current results here based on your experiment.

Reference 26; is too old you can replace it with https://doi.org/10.1016/j.sjbs.2022.01.059, or another appropriate reference.

Author Response

Dear Reviewer

I would like to thank you for the time and effort you put in evaluating this manuscript. The comments from you were very helpful. I have made the following changes and corrections (indicated in red throughout the text) on the revised manuscript.

I hope that this revised manuscript serves the purpose of your satisfaction. However, if anything else needs to be done, I would like to let you know that I am willing to cooperate. I really appreciate your help.

Comments and Suggestions for Authors

Line 37-38; (drought in 2008), please update the information as too old for now.

New reference was added

line 80; please determine the full scientific name and variety of pepper.

(Capsicum annuum) has been added. Long-fruited sweet pepper (cv. Çetinel)

line 99; how did you prepare the extraction for chemical analysis please explain?

Detailed extraction procedure has been added.

In all tables; in some values, you did not mention standard deviations, and have not shown the letter in superscripts.

Letters indicating statistical differences are shown in superscripts.  Standard deviations were not added for the sake of clarity in the tables. Instead, statistical differences were indicated by adding Duncan test. However, I can add the standard deviations if the reviewer still think otherwise.

why did you use four different controls? it was not described in the methodology.

I did not understand what the referee meant by 4 controls but I did not use 4 controls. The experimental design was 2 factorial and this information was given in the methodology (line 167). In the tables, main factors (stress and treatments) were given and their interaction effects were also given. Therefore, I had 2 different treatments for each dose of MF or US under 2 different stress conditions (drought stress or non-stress condition). However, I added a dip note to the tables and explained what the control means.

Conclusion; nothing here is about the conclusion. You just recommended for future studies, please reshape your conclusion and conclude your current results here based on your experiment

Conclusion part has been improved according to referee’s suggestions.

Reference 26; is too old you can replace it with https://doi.org/10.1016/j.sjbs.2022.01.059, or another appropriate reference.

Reference was updated

Reviewer 3 Report

The manuscript entitled “Effects of Magnetic Field and Ultrasound Applications on En- 2 dogenous Melatonin Content and Drought Stress Tolerance of 3 Pepper Seedlings” entails about the importance and efficacy of Magnetic field (MF) and ultrasound to reduce the effects of environmental stress conditions. The manuscript can be interested to wide range of scientific research in the horticulture and other related field. There are some major issues that required attentions and revision from the author prior to its final acceptance.

Abstract

Authors need to present findings as integral values or percentage advantage

Define all the abbreviated form in the abstract

Example, MDA

Introduction

Citations are missing. The author must include the citations if the statements or reports are abtained from published manuscript or other literatures.

For example

Line 28, 34. 42, 59-60.

Line 44: what are those uses? Include

“The MF has many uses.”

Line 79? What do you mean by unheated greenhouses? Explain

Line 86: Give  model, make, address (city, country) of the instruments used in the experiments

(ISOLAB),

Figure1. Keep it as supplementary data if it is really necessary, otherwise omit.

Describe the methods that author followed to perform the following experiments in detail.

MEL Content, Chlorophyll Content, Carotenoid Content, Malondialdehyde (MDA) Analysis, Total Phenolic Substance, Proline Analysis, Hydrogen peroxide (H2O2) Analysis, Enzyme Analysis

Line 168: There are many sentences formation and grammar errors. Need an extensive language editing.

Line 168: “It seen that”

Line 292: “to at varying”

Table 1,2,3,4: define *,  **,   ***   at the footnotes

Table 1,2,3,4: Statistical analysis (alphabets) are missing in some values. Maintain uniformity.

Discussion

Citations are missing

Line 275, 277, 282, 294,

Line 316: Which symptoms are you talking about? Mention it

“severe symptoms”

Line 20 and 336: Avoid repeat sentences

Major comments

Include the mechanism involve in the positive effects of MF and US applications for increasing endogenous MEL content and drought tolerance..

Discussion section is very poor. Use more recent citations to discuss the results abtained in the experiments.

Include the molecular and chemical analysis to support the results and discuss in the manuscript.

Not satisfactory. Need an extensive language editing. 

Author Response

Dear Reviewer

I would like to thank you for the time and effort you put in evaluating this manuscript. The comments from you were very helpful. I have made the following changes and corrections (indicated in red throughout the text) on the revised manuscript.

I hope that this revised manuscript serves the purpose of your satisfaction. However, if anything else needs to be done, I would like to let you know that I am willing to cooperate. I really appreciate your help.

Comments and Suggestions for Authors

The manuscript entitled “Effects of Magnetic Field and Ultrasound Applications on Endogenous Melatonin Content and Drought Stress Tolerance of 3 Pepper Seedlings” entails about the importance and efficacy of Magnetic field (MF) and ultrasound to reduce the effects of environmental stress conditions. The manuscript can be interested to wide range of scientific research in the horticulture and other related field. There are some major issues that required attentions and revision from the author prior to its final acceptance.

Abstract

Authors need to present findings as integral values or percentage advantage

I did not understand what the referee meant by integral value, however, I did not chose to give % advantage because our experimental design (factorial) is not suitable for this kind of data presentation. Therefore, I presented the actual values and we run mean separation test (Duncan) to indicate the significances on these values.

Define all the abbreviated form in the abstract Example, MDA

Abbreviations for the terms such as malondialdehyde (MDA), hydrogen peroxide (H2O2) and catalase (CAT) have been added.

Introduction

Citations are missing. The author must include the citations if the statements or reports are abtained from published manuscript or other literatures.

For example

Line 28, 34. 42, 59-60.

Missing referances added (Line 28, 34. 42). Line, 59-60 missing reference not determined. If the reviewer states it more clearly, I can check it again.

Line 44: what are those uses? Include

changed to contained

“The MF has many uses.”

Sentence removed

Line 79? What do you mean by unheated greenhouses? Explain

glass greenhouse without heating system. changed to glass greenhouses.

Line 86: Give  model, make, address (city, country) of the instruments used in the experiments

(ISOLAB),

Hall Effect equipment (5405 Dipole-GMW-California-USA) Additional information has been added.

(ISOLAB), changed to (Elma-Elmasonic S-Singen-Germany)

Figure1. Keep it as supplementary data if it is really necessary, otherwise omit.

Figure removed from the main text and added as supplementary data

Describe the methods that author followed to perform the following experiments in detail.

MEL Content, Chlorophyll Content, Carotenoid Content, Malondialdehyde (MDA) Analysis, Total Phenolic Substance, Proline Analysis, Hydrogen peroxide (H2O2) Analysis, Enzyme Analysis

Detailed information about the analyses have been added to the text

Line 168: There are many sentences formation and grammar errors. Need an extensive language editing.

Line 168: “It seen that” ???

Sentence edited.

Line 292: “to at varying”

Edited.

Table 1,2,3,4: define *,  **,   ***   at the footnotes

These definitions have been added

Table 1,2,3,4: Statistical analysis (alphabets) are missing in some values. Maintain uniformity.

I did not quite understand what the referee means but if he/she means the superscripted letters next to the means, there are not missing values. If a main factor or interaction effect is statistically significant then we run Duncan test to separate the means. If there are no letters, it means that the effect of that factor is not significant.

Discussion

Citations are missing

Line 275, 277, 282, 294, ???

Line 275, 277: Added.

Line 282: Reference [19] was added at the end of the study.

Line 294: Similarly, cold plasma (a high-frequency (40 KHz), high-voltage generator (5 kV)) applica-tion applied to mung beans increased the amount of endogenous GA3 by ~2.8 times resulting in improved seed germination and seedling root growth. The change in GA3 content was also found in parallel with radicle growth [20]. Added

Line 316: Which symptoms are you talking about? Mention it

“severe symptoms”

 Information was added.

Line 20 and 336: Avoid repeat sentences

Line 20: Sentence removed

Major comments

Include the mechanism involve in the positive effects of MF and US applications for increasing endogenous MEL content and drought tolerance..

Added.

Discussion section is very poor. Use more recent citations to discuss the results abtained in the experiments.

The Discussion part has been improved according to referee’s suggestion.

Include the molecular and chemical analysis to support the results and discuss in the manuscript.

 Added.

Reviewer 4 Report

The paper discusses the potential effects of magnetic field (MF) and ultrasound (US) applications on endogenous melatonin (MEL) levels in pepper plants and their usability in mitigating drought stress. While the study presents some interesting findings, there are several aspects that requires changes.

Firstly, the authors should add some more information to clarify the statement of the research hypothesis or objective. It would benefit from a more concise and focused research question that outlines the specific aim of the study. Without a well-defined objective, it becomes difficult to assess the significance and relevance of the results.

The abstract concludes by recommending MF and US applications as a cheap, fast, and chemical-free alternative method for increasing endogenous MEL content and drought tolerance. While this statement is interesting, it is important to acknowledge the limitations of the study, such as the lack of long-term data, potential ecological consequences, and applicability to different crop species or environmental conditions.

Moreover, the authors should add more information on the specific mechanisms through which MF and US treatments influence MEL levels and confer drought tolerance. Understanding the underlying physiological or biochemical processes is crucial for establishing the scientific rationale behind the observed effects. Without this information, it is challenging to assess the validity of the conclusions drawn from the results.

The english is of the manuscript is proper with slight minor changes in the contextus language

Author Response

Dear Reviewer

I would like to thank you for the time and effort you put in evaluating this manuscript. The comments from you were very helpful. I have made the following changes and corrections (indicated in red throughout the text) on the revised manuscript.

I hope that this revised manuscript serves the purpose of your satisfaction. However, if anything else needs to be done, I would like to let you know that I am willing to cooperate. I really appreciate your help.

Comments and Suggestions for Authors

The paper discusses the potential effects of magnetic field (MF) and ultrasound (US) applications on endogenous melatonin (MEL) levels in pepper plants and their usability in mitigating drought stress. While the study presents some interesting findings, there are several aspects that requires changes.

Firstly, the authors should add some more information to clarify the statement of the research hypothesis or objective. It would benefit from a more concise and focused research question that outlines the specific aim of the study. Without a well-defined objective, it becomes difficult to assess the significance and relevance of the results.

Since MEL is known as a broad-spectrum antioxidant that plays very important roles in plant’s response to various environmental stress factors, in this study, the possibility of whether MF applications can increase endogenous MEL levels is explored. It also investigated whether MF and US-treated seeds could induce tolerance to drought stress with a change in endogenous MEL levels. Added.

The abstract concludes by recommending MF and US applications as a cheap, fast, and chemical-free alternative method for increasing endogenous MEL content and drought tolerance. While this statement is interesting, it is important to acknowledge the limitations of the study, such as the lack of long-term data, potential ecological consequences, and applicability to different crop species or environmental conditions.

MF and US applications could be suggested as methods to increase drought tolerance in pepper by increasing endogenous MEL content. Added

Moreover, the authors should add more information on the specific mechanisms through which MF and US treatments influence MEL levels and confer drought tolerance. Understanding the underlying physiological or biochemical processes is crucial for establishing the scientific rationale behind the observed effects. Without this information, it is challenging to assess the validity of the conclusions drawn from the results.

The discussion and conclusion were arranged according to the referee's suggestion.

Comments on the Quality of English Language

The english is of the manuscript is proper with slight minor changes in the contextus language

The language of the manuscript has been improved by making small changes to the text.

Round 2

Reviewer 2 Report

.

Reviewer 3 Report

The author has revised the manuscript properly and responded to all the comments.

Now, the manuscript can be accepted in its present form.

Reviewer 4 Report

The authors replied to all my inquiries and improved the manuscript. I consider the manuscript proper to be published.